# Microglia/Macrophages in Autoimmune Demyelinating Encephalomyelitis (Multiple Sclerosis/Neuromyelitis Optica)

**DOI:** 10.3390/ijms26083585

**Published:** 2025-04-10

**Authors:** Ryo Yamasaki

**Affiliations:** Department of Neurology, Neurological Institute, Graduate School of Medical Sciences, Kyushu University, Fukuoka 812-8582, Japan; yamasaki.ryo.510@m.kyushu-u.ac.jp; Tel.: +81-92-642-5340

**Keywords:** microglia, macrophage, multiple sclerosis, neuromyelitis optica spectrum disorder, neuroinflammation, remyelination, complement inhibition, therapeutic modulation

## Abstract

Microglia and macrophages are critical mediators of immune responses in the central nervous system. Their roles range from homeostatic maintenance to the pathogenesis of autoimmune demyelinating diseases such as multiple sclerosis and neuromyelitis optica spectrum disorder. This review explores the origins of microglia and macrophages, as well as their mechanisms of activation, interactions with other neural cells, and contributions to disease progression and repair processes. It also highlights the translational relevance of insights gained from animal models and the therapeutic potential of targeting microglial and macrophage activity in multiple sclerosis and neuromyelitis optica spectrum disorder.

## 1. Introduction

Microglia and macrophages are critical mediators of central nervous system (CNS) health and disease. These cells have been studied for over a century; microglia were first identified by Pío del Río-Hortega in the 1910s as the “third element” of neural tissue, distinct from neurons and astrocytes [1]. Initially considered passive bystanders, their active roles in CNS homeostasis and immunity have gradually been elucidated. Microglia consistently increase in number in diseased CNS lesions and are believed to transform into a pro-inflammatory phenotype, similar to how peripheral monocytes often differentiate into the “M1” pro-inflammatory type. The concept of macrophage polarization emerged in the late 20th century and laid the foundation for understanding the functional diversity of these cells in both homeostatic and pathological conditions. Early categorizations, such as the binary M1/M2 framework, allowed a simplified understanding of their roles in inflammation and tissue repair but failed to capture their dynamic and context-dependent functions. Recent advances in transcriptomics and fate mapping have further refined our understanding, revealing distinct microglial and macrophage subtypes that contribute to both disease progression and repair.

Microglia are resident immune cells that are derived from yolk sac progenitors, whereas macrophages infiltrate the CNS from peripheral blood during pathological conditions. Their overlapping but distinct roles are particularly evident in autoimmune demyelinating diseases such as multiple sclerosis (MS) and neuromyelitis optica spectrum disorder (NMOSD). The present review highlights the contributions of microglia and macrophages to these diseases, emphasizing recent advances in our understanding of their mechanisms of activation, functional diversity, and therapeutic targets, as well as translational insights from animal models.

## 2. Origins and Classification of CNS Microglia and Peripheral Macrophages

### 2.1. Nomenclature of Microglia and Macrophages

The nomenclature for microglia has a complex history that has evolved substantially over the years. Initially, the binary M1/M2 classification was widely used to describe the polarized activation states of macrophages and was later extended to microglia. However, this classification oversimplifies the dynamic and multifaceted roles of these cells in various contexts, such as during development, disease progression, and aging. Recent position papers have strongly advocated that we move away from the M1/M2 framework toward a more precise functional nomenclature that incorporates specific states, molecular signatures, and contextual interactions [2,3]. For example, macrophages may be referred to as “tissue resident macrophages” to emphasize their role in homeostasis, “wound-healing macrophages” for their involvement in tissue repair, or “inflammatory macrophages” during infection. Similarly, microglia can be categorized as “surveying microglia” for their role in monitoring brain health, “reactive microglia” when responding to neuronal damage, or “disease-associated microglia” in specific pathologies such as Alzheimer’s disease [3].

### 2.2. Fate Mapping and Origins

Microglia are distinct among CNS resident immune cells because of their unique developmental origins and self-renewal capabilities. Fate mapping studies have demonstrated that microglia are derived from yolk sac progenitors during early embryogenesis, prior to the establishment of the blood–brain barrier (BBB) [4,5]. These progenitors then migrate into the developing CNS and differentiate into microglia, which persist throughout life by undergoing self-renewal within the CNS without substantial input from circulating monocytes or other hematopoietic lineages. This process distinguishes microglia from peripheral macrophages, which originate from hematopoietic precursors in the fetal liver or bone marrow and require continuous replenishment from circulating monocytes [5,6].

An essential feature of microglia is their ability to independently maintain their population within the CNS. Compared to peripheral macrophages, which are highly differentiated and exhibit robust expression of surface markers such as cluster of differentiation (CD)45, microglia are less differentiated, or microglia are relatively undifferentiated compared to peripheral macrophages, which are highly differentiated and exhibit the robust expression of surface markers such as cluster of differentiation (CD)45 [4]. The comparatively low CD45 expression levels of microglia reflect their unique regulatory environment within the CNS, which is characterized by a balance between immune surveillance and tolerance of neuronal components. This highlights their functional and developmental divergence from peripheral macrophages, which are heavily influenced by systemic immune signals.

The timing of microglial colonization within the CNS coincides with critical phases of brain development, such as neurogenesis and synaptogenesis. Microglia actively contributes to these processes by pruning synaptic connections, clearing apoptotic cells, and secreting growth factors that support neuronal maturation [7]. These activities are tightly regulated by their microenvironment, which includes interactions with astrocytes, neurons, and other glia. When the BBB has matured, microglia become physically and functionally isolated from peripheral immune cells, further emphasizing their role as the primary immune regulators within the CNS.

Recent single-cell transcriptomic studies and lineage-tracing models, such as the use of *CX3CR1-CreERT2* and *Hexb* reporter systems, have provided deeper insights into microglial self-renewal and their distinct transcriptional programs [8]. This research also revealed that microglia exhibit a high degree of plasticity; this allows them to respond dynamically to environmental changes, such as injury or disease, without losing their core identity.

In summary, microglia are uniquely adapted to the CNS environment, being both less differentiated and more self-sustaining than peripheral macrophages. Their low CD45 expression, yolk sac-derived origins, and capacity for self-renewal highlight their essential role in maintaining CNS homeostasis and responding to pathological challenges.

### 2.3. Surface Markers for Microglia and Macrophages

Microglia and macrophages are distinct yet closely related components of the CNS immune milieu. They each exhibit unique surface markers that are dependent on their activation states and steady-state conditions [2,6,9,10,11,12,13] (Table 1). In a steady state, microglia—as the CNS resident immune cells—are characterized by the expression of specific markers such as transmembrane protein 119 (TMEM119), purinergic receptor P2Y12 (P2RY12), and X3C chemokine receptor 1 (CX3CR1), which are absent from peripheral macrophages and other myeloid cells. These markers reflect the homeostatic roles of microglia, including the surveillance of the CNS environment and the maintenance of neuronal health. TMEM119 and P2RY12 are particularly noteworthy, because they are downregulated upon microglial activation, thereby distinguishing resting microglia from their reactive counterparts.

Macrophages, by contrast, are derived from peripheral monocytes or tissue resident precursors. Under homeostatic conditions, they express markers such as CD45, CD68, and CD206, which are involved in antigen presentation and tissue-specific functions (e.g., wound healing and immune surveillance). Although CD68 is commonly associated with lysosomal activity in both microglia and macrophages, the higher expression of CD45 in macrophages serves to distinguish them from their CNS resident counterparts.

Upon activation, microglia undergo phenotypic and functional transformations that are reflected in the expression of activation-associated markers. Reactive microglia upregulate major histocompatibility complex class II molecules and CD68, thereby indicating an enhanced capacity for antigen presentation and phagocytosis. Additionally, they exhibit increased expression of CD86, a costimulatory molecule that is critical for T-cell activation. This reactive state is often accompanied by the downregulation of homeostatic markers such as P2RY12, which marks the transition from a surveying to an inflammatory phenotype.

Macrophages also exhibit dynamic changes in marker expression, depending on their activation state. In pro-inflammatory (M1-like) macrophages, markers such as CD86 and inducible nitric oxide synthase are upregulated, reflecting their role in pathogen clearance and tissue destruction. Conversely, anti-inflammatory (M2-like) macrophages are characterized by the expression of CD206 and arginase-1, markers associated with tissue repair and the resolution of inflammation.

The differential expression of these markers not only provides insights into the functional states of microglia and macrophages but also underscores their distinct roles in CNS homeostasis and pathology. The ability to distinguish between these states is crucial for understanding their contributions to various neurological disorders and developing targeted therapeutic strategies. Advances in single-cell technologies and immunophenotyping continue to refine our understanding of these complex cellular dynamics and offer new opportunities for therapeutic intervention in CNS diseases.

### 2.4. Classification via Single-Cell Analysis

Advances in single-cell RNA sequencing have revealed extensive heterogeneity within macrophage populations of the CNS. For example, microglia can be distinguished from CNS-associated macrophages by their distinct gene expression patterns. CNS-associated macrophages characteristically express *Mrc1* and *Lyve1*, whereas microglia exhibit high *Hexb* and *P2ry12* levels [8]. Transgenic models, such as *Mrc1-Cre^ERT2^* mice, have been instrumental in tracing the lineage of CNS-associated macrophages back to their yolk sac progenitors, similar to microglia. These findings challenge the traditional view of separate lineages and provide a unified framework for understanding CNS macrophage development. Such insights will be crucial for elucidating the roles of CNS macrophages in CNS health and pathology.

## 3. Mechanisms of Activation and Phagocytosis Regulation in Microglia and Macrophages

Microglia and macrophages are essential components of the mononuclear phagocyte system within the CNS. They mediate immune responses, maintain homeostasis, and play dual roles in injury and repair. Their activation mechanisms, transcriptional regulation, and phagocytic functions are explored here, with an emphasis on their relevance to neurodegenerative and demyelinating diseases.

### 3.1. Activation Pathways and Molecular Signals

Microglial activation is initiated by the recognition of damage- and pathogen-associated molecular patterns via pattern recognition receptors such as Toll- and NOD-like receptors. These interactions trigger intracellular signaling cascades, including nuclear factor (NF)-κB and mitogen-activated protein kinase pathways, which induce the production of pro-inflammatory cytokines and chemokines such as tumor necrosis factor (TNF)-α, interleukin (IL)-1β, and IL-6 [14].

The colony-stimulating factor 1 receptor (CSF1R) signaling pathway is crucial for microglial survival, proliferation, and differentiation. Notably, the pharmacological inhibition of CSF1R has shown promise for depleting overactive microglia in conditions such as Alzheimer’s disease, thereby highlighting its therapeutic potential [15].

### 3.2. Transcriptional Regulation of Functional States

The activation and functional states of microglia and macrophages are tightly controlled by transcription factors including NF-κB, interferon regulatory factors, and signal transducer and activator of transcription family proteins. These regulators drive the expression of genes involved in inflammation, phagocytosis, and cell survival.

Microglia can transition from a homeostatic state to a disease-associated state under pathological conditions. This shift is characterized by the downregulation of homeostatic markers (e.g., P2RY12 and TMEM119) and the upregulation of genes associated with phagocytosis and lipid metabolism, such as triggering receptor expressed on myeloid cells 2 (TREM2) and apolipoprotein E (APOE) [10].

Similarly, macrophages exhibit functional polarization into pro-inflammatory (M1-like) and anti-inflammatory (M2-like) states. M1-like macrophages are characterized by inducible nitric oxide synthase and CD86 expression, whereas M2-like macrophages express CD206 and arginase-1, thereby promoting tissue repair and anti-inflammatory responses [13].

### 3.3. Regulation of Phagocytosis

Phagocytosis is a critical function of microglia and macrophages; it enables the clearance of apoptotic cells, myelin debris, and protein aggregates. This process is regulated by both phagocytosis-promoting and -inhibiting molecules that maintain immune homeostasis.

#### 3.3.1. Phagocytosis-Promoting Molecules

TREM2 facilitates the clearance of cellular debris and amyloid-beta, thus playing a protective role against neurodegenerative diseases such as Alzheimer’s disease [16]. In addition, CSF1R signaling supports the proliferation of microglia and enhances their phagocytic capacity; its inhibition is being explored as a therapeutic strategy for reducing chronic inflammation [15]. Finally, Fcγ receptors mediate antibody-dependent phagocytosis, thus enabling macrophages and microglia to clear opsonized pathogens and damaged cells [17].

#### 3.3.2. Phagocytosis-Inhibiting Molecules

CD33 is an immunomodulatory receptor that negatively regulates phagocytosis, thereby contributing to impaired amyloid-beta clearance and increased neurodegeneration. Targeting CD33 is a promising therapeutic approach in Alzheimer’s disease [18]. Moreover, signal regulatory protein alpha (SIRPα)/CD47 interactions provide a “do not eat me” signal that prevents excessive phagocytosis, thus protecting healthy cells. Therapeutic strategies targeting this pathway are intended to enhance the clearance of damaged or diseased cells [19].

### 3.4. Integrated Roles in Disease and Repair

The phagocytic activity of microglia and macrophages is a double-edged sword. Efficient clearance of debris is necessary for CNS repair and remyelination; however, excessive or dysregulated phagocytosis can exacerbate neuroinflammation and tissue damage. For example, in MS, although the microglial phagocytosis of myelin debris facilitates remyelination, persistent activation contributes to chronic lesion expansion [20,21].

Emerging therapies that target microglial and macrophage pathways, such as TREM2 agonists and CSF1R inhibitors, offer new opportunities to balance the reparative and pathogenic roles of these cells in CNS diseases.

## 4. Interactions Between Microglia/Macrophages and Neural Cells

### 4.1. Interactions with Neurons

Microglia continuously monitor the neuronal environment and respond to injury by altering their morphology and functional states. They phagocytose apoptotic neurons and synaptic debris, thereby facilitating synaptic pruning during development and recovery following injury. Microglia also release neurotrophic factors, such as brain-derived neurotrophic factor, which support neuronal survival and synaptic plasticity. However, chronic activation leads to the release of neurotoxic factors such as glutamate and reactive oxygen species, thus contributing to neuronal damage [3,7].

### 4.2. Interactions with Astrocytes

Astrocytes and microglia/macrophages coordinate to regulate CNS inflammation. Astrocytes release cytokines and chemokines that activate microglia and macrophages, thereby enhancing their phagocytic and immune functions. Conversely, activated microglia can modulate astrocytic reactivity, which influences glial scar formation in response to CNS injuries. In NMOSD, astrocytic damage caused by aquaporin 4 (AQP4)-immunoglobulin (Ig)G further amplifies microglial activation, thus perpetuating inflammatory cycles [22].

### 4.3. Interactions with Oligodendrocytes

Microglia play a dual role in oligodendrocyte function. On the one hand, they promote remyelination by clearing myelin debris and secreting growth factors such as fibroblast growth factor and platelet-derived growth factor. On the other hand, excessive inflammation driven by microglia can impair oligodendrocyte progenitor cell maturation and remyelination. Targeting microglial polarization to favor anti-inflammatory states has therapeutic potential for demyelinating diseases such as MS [23].

### 4.4. Modulatory Effects on Neural Cell Functions

Microglia-derived cytokines influence neuronal gene expression, synaptic transmission, and plasticity, thereby affecting cognitive processes such as learning and memory. Similarly, interactions between astrocytes and microglia regulate neurotrophic factor release and neurotransmitter uptake, which further modulates CNS homeostasis. The inflammation-induced disruption of these processes can lead to widespread neural dysfunction [24].

### 4.5. Microglia and Connexins (Cx)

Cx such as Cx43 form gap junctions and hemichannels in microglia and astrocytes, thus facilitating intercellular communication. Cx also mediate calcium wave propagation and inflammatory mediator release, thereby influencing CNS inflammation and repair. The finding that dysregulated Cx activity exacerbates neuroinflammation highlights the potential of Cx as therapeutic targets for modulating microglial and astrocytic responses in neurodegenerative diseases [25,26].

## 5. Multiple Sclerosis: Disease Mechanisms and Microglia/Macrophage Involvement

### 5.1. Disease Overview

MS is a chronic inflammatory demyelinating disease of the CNS that affects approximately 2.5 million individuals worldwide. It is most common in Western countries, with prevalence rates ranging from 30 to 150 per 100,000 people; much lower rates are observed in Asia and Africa. Notably, Japan has experienced an increase in prevalence from 1.4 per 100,000 in the 1980s to 7.7 per 100,000 in recent decades. MS typically presents in young adults aged 20–40 years, with women affected approximately three times more often than men (Figure 1).

The pathogenesis of MS involves an autoimmune response in which autoreactive T cells infiltrate the CNS and target myelin antigens, causing inflammation and demyelination. This process is driven by cytokines, such as interferon-gamma (IFN-γ) and IL-17, alongside the activation of B cells and macrophages. In progressive stages, inflammation becomes compartmentalized within the CNS and is dominated by activated microglia and macrophages.

Genetic susceptibility plays an important role in MS; the *HLA-DRB1*15:01* allele is the most prominent risk factor. Environmental triggers, including Epstein–Barr virus infection, low vitamin D levels, and smoking, further contribute to disease risk, interacting with genetic factors in a complex manner.

Clinically, MS is characterized by a relapsing–remitting course in most patients, with approximately 50% transitioning to secondary progressive MS over time (Figure 1). A smaller proportion present with primary progressive MS, which is marked by steady neurological decline from onset. Although disease-modifying therapies have markedly reduced relapse rates in relapsing–remitting MS, effective treatments for primary and secondary progressive MS remain limited. Addressing neurodegeneration, inflammation of the CNS, and the promotion of remyelination are critical unmet needs in MS management [20].

### 5.2. Animal Models of MS

Animal models are indispensable for understanding the complex pathophysiology of MS and developing new therapeutic strategies. These models replicate specific features of MS, such as neuroinflammation, demyelination, and neurodegeneration, and can be broadly categorized into immunological, toxin-induced, viral, and transgenic models (Table 2).

#### 5.2.1. Experimental Autoimmune Encephalomyelitis (EAE)

EAE is the most widely used immunological model of MS; it is induced by active immunization with myelin-derived proteins (e.g., myelin basic protein, proteolipid protein [PLP], or myelin oligodendrocyte glycoprotein [MOG]) or the passive transfer of activated CD4^+^ T lymphocytes. EAE mimics MS-like features, including CNS inflammation, demyelination, and axonal loss. Variants of EAE models have been used to replicate different forms of MS, for example, relapsing–remitting MS can be modeled by immunization with PLP_139–151_ in Swiss Jim Lambert mice, whereas chronic progressive MS can be simulated using MOG_35–55_ immunization in C57BL/6 mice. However, EAE has limitations, such as its overreliance on T-cell-mediated pathology, which does not fully reflect the broader immune mechanisms of human MS [27].

#### 5.2.2. Toxin-Induced Demyelination Models

Toxins such as cuprizone, lysolecithin, and ethidium bromide can be used to induce demyelination and study the mechanisms behind its repair. In the cuprizone model, oral cuprizone administration induces oligodendrocyte apoptosis and demyelination, particularly in the corpus callosum. This model is effective for studying remyelination and glial responses but lacks substantial inflammatory components [28]. Meanwhile, in the lysolecithin and ethidium bromide models, these agents are directly injected into specific CNS regions to create focal demyelination. These models are valuable for studying localized demyelination and subsequent remyelination [29].

#### 5.2.3. Viral Models

Theiler’s murine encephalomyelitis virus infection induces progressive demyelination that resembles the chronic stages of MS. This model effectively mimics the involvement of CD8^+^ T cells and B cells, which are critical in MS pathology. However, its use is restricted to specific mouse strains that are susceptible to viral infection [30].

#### 5.2.4. Transgenic Models

Transgenic models allow the investigation of specific molecular and cellular mechanisms underlying MS. For example, MOG-specific T-cell receptor transgenic mice spontaneously develop MS-like features, thereby enabling the investigation of T-cell-mediated demyelination [31]. Similarly, oligodendrocyte-specific Cx-knockout mice have been used to model progressive forms of MS, thus highlighting the role of Cx in maintaining myelin integrity and in disease progression [35].

#### 5.2.5. Strengths and Limitations

Each animal model like those mentioned above offers unique insights but has inherent limitations. Specifically, EAE is robust for modeling immune-mediated demyelination but lacks MS features such as remyelination efficiency and chronic progression. Toxin-induced models are excellent for studying remyelination but do not capture the autoimmune aspects of MS. Viral and transgenic models address specific pathological components but require specialized settings and often fail to replicate the full spectrum of MS [36].

### 5.3. Roles of Microglia and Macrophages in MS Pathogenesis

MS is characterized by chronic inflammation, demyelination, and neurodegeneration in the CNS. Microglia (the resident immune cells of the CNS) and macrophages (both resident and peripherally infiltrating) are critical mediators of MS pathogenesis. These cells contribute to both tissue damage and repair, reflecting their dual roles in CNS homeostasis and immune response (Figure 2).

#### 5.3.1. Contribution to Inflammation and Demyelination

Microglia and macrophages are early responders to CNS injury in MS. Activated microglia adopt an amoeboid shape and upregulate pro-inflammatory mediators such as IL-1β, TNF-α, and nitric oxide, which exacerbate tissue damage by promoting oxidative stress and excitotoxicity. These cells also produce matrix metalloproteinases (MMPs), leading to BBB disruption and facilitating immune cell infiltration. Peripherally derived macrophages infiltrate the CNS during acute lesions and amplify the inflammatory response by presenting antigens to T cells and releasing pro-inflammatory cytokines [37].

#### 5.3.2. Chronic Lesion Dynamics

In chronic MS lesions, microglia and macrophages cluster at the lesion borders; this contributes to slow lesion expansion and sustained neuroinflammation. Studies have identified distinct transcriptional profiles for microglia and macrophages in active lesion borders versus quiescent areas [20]. For example, iron-laden microglia in chronic lesions exhibit the upregulation of genes related to iron metabolism and immune activation, such as TREM2 and APOE, which are implicated in lesion persistence [38].

#### 5.3.3. Neurodegenerative Processes

During progressive MS, compartmentalized inflammation within the CNS drives neurodegeneration. Microglia in this phase exhibit reduced homeostatic gene expression (e.g., P2RY12) and increased inflammatory marker expression (e.g., CX3CR1). These reactive states impair remyelination by failing to effectively clear myelin debris and by producing neurotoxic molecules, such as glutamate and reactive oxygen species [39].

#### 5.3.4. Role in Repair and Remyelination

Despite their pathogenic roles, microglia and macrophages are crucial for tissue repair. They clear myelin debris, secrete growth factors such as insulin-like growth factor 1, and promote oligodendrocyte precursor cell differentiation. Pro-repair phenotypes, often referred to as M2-like states, dominate in resolving lesions. However, the transition from pro-inflammatory (M1-like) to reparative (M2-like) states is frequently impaired in MS, which may limit recovery [40].

#### 5.3.5. Emerging Therapeutic Implications

Targeting microglial and macrophage pathways offers promising therapeutic avenues. For example, TREM2 agonists enhance myelin debris clearance and remyelination, whereas CSF1R inhibitors selectively deplete overactive microglia to reduce neuroinflammation. Future therapies should aim to manipulate the polarization of microglia and macrophages to promote their reparative functions and mitigate their neurotoxic effects [41,42].

### 5.4. Mechanisms of Action of MS Therapies Through Microglia/Macrophage Modulation

MS treatment has evolved to target key mechanisms of the immune response, including those involving microglia and macrophages. Although most therapies aim to modulate peripheral immune activity, an increasing number of strategies focus on CNS resident immune cells to address both the relapsing and progressive forms of MS (Table 3).

#### 5.4.1. Anti-Inflammatory Therapies

Bruton tyrosine kinase inhibitors (BTKIs) represent a novel class of MS therapeutic agents that target CNS resident microglia and peripheral myeloid cells. By inhibiting BTK, they suppress intracellular signaling involved in B-cell receptor activation and microglial cytokine production. CNS-penetrant BTKIs, such as tolebrutinib and orelabrutinib, have demonstrated efficacy for reducing lesion formation in relapsing MS and are under investigation for progressive MS forms. Their potential to modulate microglial activation and limit chronic neuroinflammation makes them a promising avenue for addressing progressive MS [43].

Drugs such as fingolimod and siponimod modulate sphingosine-1-phosphate (S1P) receptors to sequester lymphocytes in lymphoid tissues, thus reducing their CNS infiltration. Additionally, S1P receptor signaling in microglia influences their activation state and cytokine production, thereby reducing CNS inflammation and promoting a neuroprotective environment [44].

Monoclonal antibody therapies, such as anti-CD20 therapies (e.g., ocrelizumab and ofatumumab), deplete B cells, thus indirectly reducing their activation of microglia via cytokine signaling. Similarly, natalizumab (an integrin antagonist) prevents immune cell migration across the BBB, which decreases macrophage-mediated tissue damage in active lesions [45].

#### 5.4.2. Remyelination-Promoting Strategies

Originally identified as a remyelination agent, clemastine fumarate promotes oligodendrocyte progenitor cell differentiation and myelin repair. Although it does not directly target microglia, its effects are complemented by microglial phagocytosis of myelin debris, which is a prerequisite for effective remyelination [46].

High-dose biotin (MD-1003) enhances fatty acid synthesis and axonal energy metabolism, thereby indirectly supporting remyelination. The effects of biotin on microglia include promoting an anti-inflammatory and reparative state, which is critical for creating an environment that is conducive to myelin repair [47].

#### 5.4.3. Neuroprotection and Microglial Modulation

Masitinib is a selective tyrosine kinase inhibitor that targets microglia and mast cells to reduce chronic CNS inflammation. Recent phase III trials have demonstrated its potential to slow disease progression in primary and secondary progressive MS by suppressing microglial activation [48].

#### 5.4.4. Emerging Therapies

Autologous Epstein–Barr virus-reactive T cells and antiviral agents targeting latent Epstein–Barr virus infection are under investigation for their ability to reduce B-cell-mediated microglial activation. This approach is particularly relevant for addressing the neuroinflammatory mechanisms implicated in progressive MS [49].

Therapies aimed at modifying gut microbiota, such as fecal microbiota transplantation and bile acid derivatives (e.g., tauroursodeoxycholic acid), indirectly influence microglial activation via systemic immune modulation. This strategy highlights the role of systemic inflammation in CNS pathophysiology [50].

#### 5.4.5. Limitations and Future Directions

Although many therapies effectively reduce inflammation and disease activity, most target peripheral immune processes rather than the CNS resident microglia. Progressive MS, characterized by chronic microglial activation, remains challenging to treat. Novel approaches, including CNS-penetrating agents and therapies targeting remyelination and neuroprotection, are essential for improving MS management.

## 6. Neuromyelitis Optica: Disease Mechanisms and Microglia/Macrophage Involvement

### 6.1. Disease Overview

NMOSD is a rare autoimmune disease characterized by multifocal CNS inflammation that primarily affects the optic nerves, spinal cord, and specific brain regions. Historically known as Devic’s disease, NMOSD was once considered a severe variant of MS. However, the discovery of AQP4-IgG autoantibodies identified NMOSD as a separate disease entity with a chronic, relapsing course [51].

NMOSD accounts for approximately 1–2% cases of CNS inflammatory demyelinating diseases in Western populations; it is more prevalent in Asian, African, and Afro-Caribbean populations, where it constitutes up to one-third of such cases. Global prevalence estimates range from 0.7 to 10 cases per 100,000 individuals. The median age of onset is 40 years, although it can occur at any age. Women are disproportionately affected, particularly in AQP4-IgG-positive cases, in which the female-to-male ratio approaches 9:1.

AQP4-IgG autoantibodies play a central role in NMOSD pathogenesis by targeting AQP4 water channels expressed on astrocytic endfeet in the BBB. Antibody binding initiates complement activation, which leads to astrocytic damage, inflammatory lesion formation, and secondary demyelination. This mechanism distinguishes NMOSD from MS, in which oligodendrocytic damage occurs first.

NMOSD is characterized by six core clinical syndromes: (1) optic neuritis (unilateral or bilateral vision loss, often accompanied by pain with eye movement); (2) acute myelitis (longitudinally extensive transverse myelitis causing motor, sensory, and autonomic dysfunction); (3) area postrema syndrome (persistent hiccups, nausea, and vomiting); (4) brainstem syndromes (symptoms such as vertigo, dysarthria, and cranial nerve deficits); (5) diencephalic syndromes (disorders such as narcolepsy, endocrine dysfunctions, and eating disorders); and (6) cerebral syndromes (rare presentations with encephalopathy, seizures, or focal neurological deficits).

AQP4-IgG antibodies are detectable in >80% of NMOSD cases. AQP4-IgG-negative cases include patients with antibodies against MOG-IgG or other, as yet unidentified, autoantibodies.

A NMOSD diagnosis requires one or more core clinical syndromes, AQP4-IgG antibody detection, and the exclusion of alternative diagnoses. Advanced imaging techniques such as magnetic resonance imaging reveal characteristic longitudinally extensive lesions in the spinal cord or specific brain regions, which can support a clinical diagnosis.

NMOSD follows a relapsing course in >90% of cases. Attacks typically peak within days, often leaving permanent neurological deficits. Without treatment, the cumulative disability from recurrent attacks can lead to marked morbidity, including blindness, paraplegia, and even death in severe cases (Figure 1). The early diagnosis and treatment of NMOSD are critical for preventing relapses and preserving neurological function [32,52].

### 6.2. Animal Models of NMOSD

Animal models are essential for elucidating the mechanisms underlying NMOSD and testing potential therapeutic interventions. The defining characteristic of NMOSD models is the focus on AQP4-IgG autoantibodies and their effects on astrocytes, which distinguish NMOSD from other demyelinating diseases such as MS (Table 2).

#### 6.2.1. Passive Transfer Models

In passive transfer models, AQP4-IgG purified from NMOSD patients is administered to animals. These models rely on disrupting the BBB to allow the antibodies to access astrocytes in the CNS. For example, in EAE-based models, animals are first induced to develop EAE using myelin-related antigens. AQP4-IgG is then injected to replicate the astrocytopathy observed in NMOSD. These models have astrocytic damage, complement activation, and inflammatory cell infiltration. In models established by intrathecal or intracerebral administration, AQP4-IgG and human complement are directly injected into the CNS, bypassing the BBB. These models show astrocyte injury, loss of AQP4 and glial fibrillary acidic protein, and complement deposition within days [33].

#### 6.2.2. Active Immunization Models

Active immunization with AQP4 peptides has been used to study the contributions of T cells to NMOSD pathology. Such models feature robust immune responses that mimic chronic inflammation. T cells activated against AQP4 exacerbate astrocytic damage and increase lesion severity [34]; however, these models are limited by their inability to fully replicate AQP4-IgG-mediated pathology.

#### 6.2.3. Limitations of Current Models

There are species-specific challenges when attempting to establish the currently available models. For example, mouse complement does not efficiently interact with human AQP4-IgG, and supplementation with human complement or that of an alternative animal species (e.g., rats) is required for successful model establishment. Moreover, many models rely on artificial BBB disruption, which does not fully mimic spontaneous NMOSD onset in humans. Additionally, although these models replicate astrocytic damage and inflammatory infiltration, they often lack the characteristic features of human NMOSD, such as chronic relapses and remissions.

#### 6.2.4. Advances in Model Development

Recent studies have introduced genetically modified animals such as CD59-deficient mice, which lack complement regulation and are more susceptible to AQP4-IgG-mediated astrocytopathy. These advances enhance the fidelity of NMOSD models and facilitate the evaluation of complement-targeted therapies [53].

### 6.3. Roles of Microglia and Macrophages in NMOSD Pathogenesis

NMOSD is characterized by a distinct immunopathological process that involves extensive astrocytic damage and secondary demyelination. Microglia and macrophages—key immune effector cells in the CNS—play critical roles in this pathology by contributing to astrocytopathy, neuroinflammation, and tissue damage (Figure 2).

#### 6.3.1. Microglial Activation in NMOSD

Microglia are the resident immune cells of the CNS and are among the first responders to astrocytic injury triggered by AQP4-IgG binding. In NMOSD, activated microglia exhibit amoeboid morphology and upregulated CD68 and TMEM119, indicating their involvement in inflammatory responses. These cells also release pro-inflammatory cytokines such as IL-1β, TNF-α, and IL-6, thus exacerbating astrocytic damage and promoting peripheral immune cell recruitment [54].

#### 6.3.2. Macrophage Recruitment and Activation

Infiltrating macrophages contribute to NMOSD pathology. During acute NMOSD attacks, CD163- and CD14-expressing macrophages are markedly increased in number in cerebrospinal fluid and CNS lesions. These cells phagocytose cellular debris and release additional inflammatory mediators, thus amplifying local inflammation. Unlike microglia, macrophages are predominantly derived from peripheral monocytes and are recruited into the CNS through chemokine signaling pathways that involve monocyte chemoattractant protein-1 (MCP-1) and other factors [55].

#### 6.3.3. Complement-Mediated Effects

The complement system plays a central role in NMOSD pathogenesis by mediating astrocytic cytotoxicity and amplifying inflammation. Microglia and macrophages contribute to this process by producing complement components such as C1q and C3, which exacerbate astrocytic damage and further activate these immune cells in a positive feedback loop [56].

### 6.4. Therapeutic Agents for NMOSD and Their Effects on Microglia and Macrophages

Advances in therapeutic strategies against NMOSD have led to therapies that target immune mechanisms, including via microglial and macrophage modulation. These therapeutic agents aim to reduce inflammation and prevent astrocytic damage by altering the activation and function of these immune cells [57] (Table 3).

#### 6.4.1. Complement Inhibition

Eculizumab is a monoclonal antibody targeting complement protein C5; it inhibits the formation of the membrane attack complex, thereby preventing complement-mediated astrocytic injury. This reduces microglial activation and macrophage recruitment, which are driven by complement components such as C3a and C5a. By blocking complement activation, eculizumab reduces neuroinflammation and tissue damage [58].

#### 6.4.2. IL-6 Receptor Blockade

Satralizumab and tocilizumab target the IL-6 receptor, a cytokine receptor that is heavily involved in NMOSD pathogenesis. By inhibiting IL-6 signaling, these agents suppress the pro-inflammatory activation of microglia and macrophages. IL-6-mediated pathways promote the microglial secretion of inflammatory mediators and macrophage recruitment to lesions, contributing to the inflammatory cascade [59].

#### 6.4.3. B-Cell Depletion Therapies

Inebilizumab and rituximab are monoclonal antibodies against CD19 and CD20, respectively. They deplete B cells and reduce AQP4-IgG production, which indirectly decreases microglial activation, because the presence of AQP4-IgG drives microglial-mediated inflammation through Fc receptor signaling. Additionally, macrophage activation via antibody-dependent cellular cytotoxicity is mitigated [60].

#### 6.4.4. Intravenous (IV)Ig Therapy

IVIg is used off-label in NMOSD for its immunomodulatory effects. It suppresses pro-inflammatory responses by (1) inhibiting Fc receptor-mediated activation of microglia and macrophages, thus reducing cytokine release and inflammation; (2) modulating complement activation, thereby limiting complement-mediated astrocytic damage; and (3) promoting anti-inflammatory cytokine production, which helps to shift macrophages to a reparative (M2-like) phenotype. Although not a first-line therapy, IVIg is often considered in cases where standard treatments fail or are contraindicated [61].

#### 6.4.5. Steroid Pulse Therapy

High-dose methylprednisolone (steroid pulse therapy) is a cornerstone treatment for acute NMOSD relapses. It suppresses microglial and macrophage activation by reducing the production of pro-inflammatory cytokines (e.g., TNF-α and IL-1β). Moreover, it downregulates antigen presentation and Fc receptor expression, thereby dampening immune responses. It also promotes BBB stabilization, thus preventing the further infiltration of peripheral immune cells into the CNS. However, although it is effective for acute symptom management, the long-term use of steroids is limited by their side effects [62].

#### 6.4.6. Plasma Exchange

Plasma exchange is a pivotal therapeutic option for acute NMOSD relapses, particularly in cases refractory to high-dose corticosteroids. By removing circulating autoantibodies, including AQP4-IgG, plasma exchange reduces complement activation and the downstream inflammatory cascade. This approach indirectly mitigates microglial and macrophage activation, thus lowering the pro-inflammatory cytokine levels and complement-mediated astrocytic damage. Plasma exchange is often used as a bridge therapy when transitioning patients to long-term immunosuppressive regimens [63].

#### 6.4.7. Emerging Therapeutic Targets

CSF1R signaling regulates microglial survival and activation. Emerging therapies targeting CSF1R inhibition aim to modulate microglial proliferation and activity, thereby potentially reducing NMOSD-associated inflammation [64]. Additionally, by targeting the “do not eat me” signal between SIRPα and CD47, agents that target the SIRPα/CD47 axis may enhance the phagocytosis of debris and support tissue repair by macrophages [65].

#### 6.4.8. Mechanistic Insights

The aforementioned therapies not only prevent acute inflammation but also promote an environment conducive to repair. For example, IL-6 blockade fosters a shift toward anti-inflammatory macrophage phenotypes, whereas complement inhibition reduces chemotactic signals that would otherwise exacerbate inflammation. IVIg and steroids further contribute to immune modulation by targeting both systemic and CNS resident immune cells, thus ensuring a multifaceted approach to NMOSD management.

## 7. Similarities and Differences in Microglial and Macrophage Actions in MS and NMOSD

Microglia and macrophages are key immune effector cells in the CNS. They play pivotal roles in the pathogenesis and repair of MS and NMOSD. Although they have overlapping functions in these diseases, their specific roles diverge because of the distinct immunopathological processes underlying MS and NMOSD. This section outlines the similarities and differences in microglial and macrophage actions in these conditions (Figure 2).

### 7.1. Similarities

#### 7.1.1. Pro-Inflammatory Activation

In both MS and NMOSD, microglia and macrophages adopt pro-inflammatory phenotypes during active disease phases. These cells secrete cytokines such as IL-1β, TNF-α, and IL-6, which amplify neuroinflammation. Furthermore, the activation of pattern recognition receptors, such as Toll-like receptors, drives these inflammatory responses in both diseases.

#### 7.1.2. BBB Disruption

Microglia and macrophages play a pivotal role in the disruption of the BBB during neuroinflammation. Activated microglia secrete matrix metalloproteinases (particularly MMP-9), which degrade tight junction proteins such as occludin and claudin-5 in the endothelial layer, thereby increasing BBB permeability. In addition, microglia produce reactive oxygen and nitrogen species (ROS and NO), which cause oxidative stress and further damage to endothelial cells. These changes weaken the integrity of the BBB and facilitate the infiltration of circulating immune cells. Recruited macrophages amplify BBB disruption through the release of vascular endothelial growth factor (VEGF) and pro-inflammatory cytokines such as TNF-α and IL-1β. These molecules promote endothelial permeability and destabilize the neurovascular unit. Together, these processes create a permissive environment for the entry of autoreactive lymphocytes and myeloid cells into the CNS, sustaining chronic inflammation and lesion formation in MS [66].

#### 7.1.3. Phagocytosis and Debris Clearance

Microglia and macrophages are involved in phagocytosing myelin debris and cellular remnants, which is a critical step in tissue repair and remyelination. Pro-repair (M2-like) phenotypes are observed in resolving lesions of both diseases.

#### 7.1.4. Complement Activation

Complement components such as C1q and C3 are produced by microglia and macrophages in MS and NMOSD, thus exacerbating inflammation and tissue damage.

#### 7.1.5. Dual Role in Pathogenesis and Repair

Both microglia and macrophages exhibit a dual role: they mediate tissue damage during active inflammation and support repair through growth factor secretion and debris clearance.

### 7.2. Differences

#### 7.2.1. Primary Targets of Damage

In MS, microglia and macrophages primarily target oligodendrocytes and myelin, leading to demyelination and axonal damage. In NMOSD, astrocytes are the primary targets because of the pathogenic effects of AQP4-IgG autoantibodies; demyelination is a secondary consequence.

#### 7.2.2. Role of Complement

Complement-mediated cytotoxicity is more prominent in NMOSD; the binding of AQP4-IgG to astrocytes triggers complement activation and astrocytic lysis. In MS, complement activation contributes to demyelination but is not primarily driven by a specific autoantibody.

#### 7.2.3. Inflammatory Profile

In MS, microglia and macrophages display chronic inflammatory states; compartmentalized inflammation in progressive stages is driven by iron-laden microglia at lesion borders. In NMOSD, inflammation is more acute and episodic and coincides with relapses triggered by AQP4-IgG-mediated astrocytopathy.

#### 7.2.4. Recruitment of Peripheral Macrophages

The recruitment of peripheral monocytes and their differentiation into macrophages represents a key immunopathological mechanism in both MS and NMOSD, albeit through distinct pathways and with differing consequences.

In MS, peripheral monocytes are gradually recruited across a disrupted BBB, particularly during active lesion formation. Chemokines such as CCL2 (MCP-1), CCL5 (RANTES), and CXCL10 are upregulated in inflamed CNS tissue, attracting CCR2⁺ and CX3CR1⁺ monocytes from the circulation [67]. Once infiltrated, these monocytes differentiate into macrophages with either a M1-like (pro-inflammatory) or a M2-like (anti-inflammatory) phenotype, depending on the local cytokine milieu. M1-like macrophages release TNF-α, IL-6, and nitric oxide (NO), contributing to sustained inflammation, whereas M2-like macrophages secrete IL-10 and TGF-β, facilitating the clearance of debris and remyelination [68].

In neuromyelitis optica spectrum disorder (NMOSD), macrophage infiltration occurs more acutely and prominently during relapse episodes. Astrocyte damage mediated by AQP4-IgG and complement activation leads to the rapid upregulation of CCL2 and C5a in the perivascular space, both of which are strong chemoattractants for CCR2⁺ monocytes [69]. These monocytes differentiate into CD163⁺ and CD14⁺ macrophages within lesions, where they release IL-6 and TNF-α, further exacerbating astrocytic injury and promoting lesion expansion. Unlike in MS, the acute-phase lesions in NMOSD show minimal evidence of reparative macrophage activity [69].

#### 7.2.5. Disease-Specific Phenotypes

In MS, disease-associated microglia are characterized by transcriptional profiles that promote phagocytosis and inflammation. Disease-associated microglia are less clearly defined in NMOSD. NMOSD lesions exhibit higher densities of CD163^+^ macrophages, which are linked to complement-mediated damage and repair processes specific to astrocytopathy.

#### 7.2.6. Therapeutic Implications

In MS, therapies targeting microglial activation (e.g., CSF1R inhibitors and TREM2 agonists) are intended to modulate chronic inflammation and enhance remyelination. In NMOSD, complement inhibitors (e.g., eculizumab) and IL-6 receptor blockers (e.g., satralizumab) are more effective because of the central role of AQP4-IgG and complement-mediated astrocytic damage in this disease.

### 7.3. Clinical Implications

Understanding the similarities and differences in microglial and macrophage actions in MS and NMOSD provides a framework for developing targeted therapies. By tailoring treatments to the specific immunopathological mechanisms of each disease, it is possible to improve the efficacy and reduce side effects. For example, for MS, strategies focusing on remyelination and chronic microglial modulation are essential for addressing progressive MS stages. For NMOSD, early intervention with complement inhibitors and antibody-depleting therapies is critical for preventing astrocytic damage and relapses.

## 8. Conclusions

Microglia and macrophages play dual roles in CNS autoimmune diseases, driving both tissue damage and repair. Advances in single-cell analysis and fate mapping have refined our understanding of their origins and functional diversity. Targeted therapies that modulate these cells—such as CSF1R inhibitors, IL-6 receptor antagonists, and complement inhibitors—offer promising avenues for treating MS and NMOSD. Future research should focus on tailoring these approaches to enhance repair mechanisms while minimizing neuroinflammation, thereby improving patient outcomes.

## Figures and Tables

**Figure 1 ijms-26-03585-f001:**
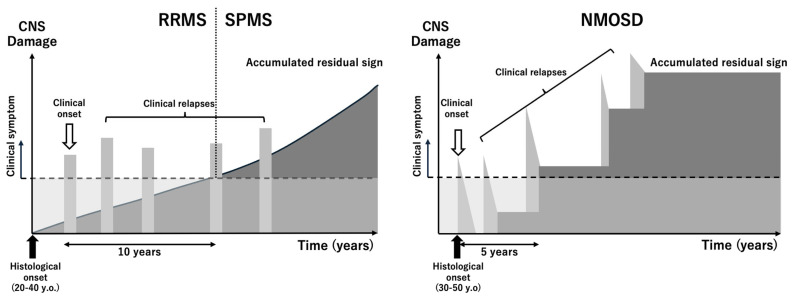
Distinct clinical and pathological courses of multiple sclerosis (MS) and neuromyelitis optica spectrum disorder (NMOSD). **Multiple sclerosis (MS)**: MS is characterized by two major pathogenic components: inflammation and neurodegeneration. Histological changes may begin decades before clinical onset, typically between 20 and 40 years of age. The disease usually starts with a relapsing–remitting course (RRMS), followed by a secondary progressive phase (SPMS), during which neurodegeneration progresses independently of relapses. This progression is inconsistent with the concept that inflammation alone drives accumulating disability. Instead, chronic neurodegenerative processes, including mitochondrial dysfunction, oxidative stress, and cortical demyelination, become dominant in later stages, leading to the gradual accrual of irreversible deficits. **Neuromyelitis optica spectrum disorder (NMOSD)**: In contrast, NMOSD pathology is largely relapse-dependent. Clinical and pathological changes typically begin between 30 and 50 years of age and coincide closely in time. Neurodegeneration in NMOSD is secondary to acute inflammatory attacks, primarily mediated by aquaporin-4 (AQP4) antibody-induced astrocyte destruction. Unlike MS, a progressive phase independent of relapses is extremely rare in NMOSD. Residual disability accumulates with each relapse, but there is little evidence of ongoing neurodegeneration between attacks, reflecting the distinct pathophysiological mechanisms underlying the disease.

**Figure 2 ijms-26-03585-f002:**
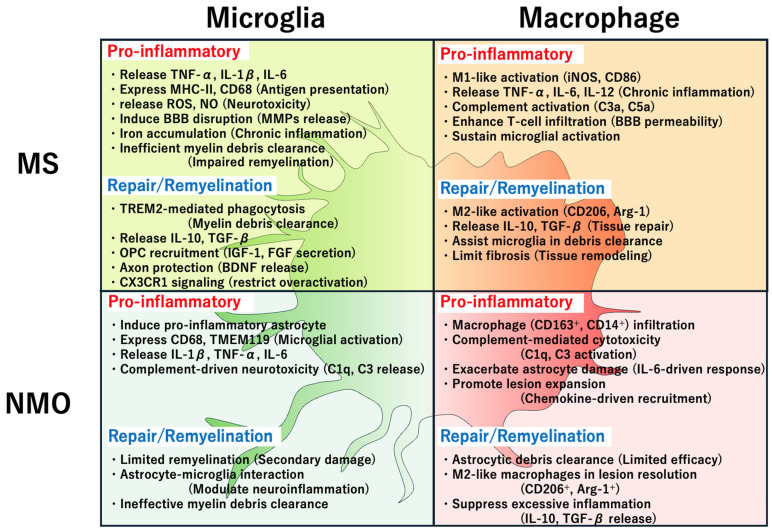
Roles of microglia and macrophages in MS and NMOSD. This figure compares the inflammatory and reparative functions of microglia and macrophages in multiple sclerosis (MS; **left**) and neuromyelitis optica spectrum disorder (NMOSD; **right**). Each quadrant illustrates disease-specific actions of CNS resident microglia and peripherally derived macrophages. The green-shaded background represents microglial functions, whereas the red-shaded background represents macrophage functions, visually separating their respective roles. Within each quadrant, red text denotes pro-inflammatory actions, including cytokine release (e.g., TNF-α, IL-1β, and IL-6), complement activation; matrix metalloproteinase (MMP)-mediated BBB disruption; and oxidative stress. Meanwhile, blue text highlights reparative or anti-inflammatory functions, such as M2-like polarization (e.g., CD206⁺ and Arg-1⁺); TREM2-mediated phagocytosis; growth factor release (e.g., IGF-1, BDNF, and TGF-β); and tissue remodeling. Key surface markers (e.g., CD68, TMEM119, CD86, and CD206) and mediators (e.g., ROS, NO, and complement components) are included to clarify the immunological mechanisms involved. This figure underscores both shared and distinct contributions of microglia and macrophages to CNS pathology and repair in MS and NMOSD.

**Table 1 ijms-26-03585-t001:** Markers for the microglia and macrophages.

Cell Type	State	Surface Markers	Features/Functions
Microglia	Steady state	TMEM119, P2RY12, CX3CR1	Surveillance of the CNS, neuroprotection, and synaptic remodeling
	Activated state	MHC class II, CD68, CD86	Antigen presentation, secretion of pro-inflammatory cytokines, neurotoxicity
Macrophages	Steady state	CD45 (high expression), CD68, CD206	Antigen presentation, immune surveillance, and tissue repair
	Activated state	M1-like: CD86, iNOS	Pro-inflammatory, pathogen clearance, tissue destruction
		M2-like: CD206, Arginase-1	Anti-inflammatory, tissue repair, resolution of inflammation

Abbreviations: CD, cluster of differentiation; CNS, central nervous system; CX3CR1, CX3C chemokine receptor 1; iNOS, inducible nitric oxide synthase; MHC, major histocompatibility complex; P2RY12, purinergic receptor P2Y12; TMEM119, transmembrane protein 119.

**Table 2 ijms-26-03585-t002:** Disease models of MS/NMO.

Disease	Model Name	Key Features	References
MS	EAE (Experimental Autoimmune Encephalomyelitis)	T cell-mediated inflammation and demyelination; most widely used MS model	[27]
	Cuprizone Model	Oligodendrocyte toxicity-induced demyelination; models remyelination	[28]
	Lysolecithin Model	Localized demyelination via lipid disruption; used for focal lesion studies	[29]
	TMEV (Theiler’s Murine Encephalomyelitis Virus) Model	Virus-induced chronic demyelination; models progressive MS	[30]
	Transgenic MOG-TCR Model	Spontaneous MS-like autoimmunity; models T cell responses	[31]
NMO	Passive Transfer Model	AQP4-IgG transfer induces astrocyte injury; requires BBB disruption	[32]
	Active Immunization Model	AQP4 peptide immunization mimics chronic inflammation	[33]
	Intrathecal AQP4-IgG Model	Direct administration of AQP4-IgG and complement; astrocytopathy model	[34]
	CD59-Knockout Model	Complement regulatory deficiency increases AQP4-IgG pathology	[34]

Abbreviations: AQP4, aquaporin 4; BBB, blood–brain barrier; CD, cluster of differentiation; EAE, experimental autoimmune encephalomyelitis; IgG, immunoglobulin G; MOG, myelin oligodendrocyte glycoprotein; MS, multiple sclerosis; NMO, neuromyelitis optica; TCR, T-cell receptor; TMEV, Theiler’s murine encephalomyelitis virus.

**Table 3 ijms-26-03585-t003:** Therapeutic approaches for MS/NMO.

Disease	Drug	Mechanism of Action
MS	Ocrelizumab	Anti-CD20 monoclonal antibody; B-cell depletion
	Natalizumab	Anti-α4 integrin monoclonal antibody; inhibits immune cell migration
	Fingolimod	S1P receptor modulator; prevents lymphocyte egress
	Siponimod	Selective S1P1/S1P5 modulator; reduces neuroinflammation
	Tolebrutinib	BTK inhibitor; suppresses microglial activation
	Clemastine fumarate	Promotes OPC differentiation and remyelination
	Masitinib	Tyrosine kinase inhibitor; reduces microglial activation
NMO	Eculizumab	C5 complement inhibitor; prevents complement-mediated astrocyte injury
	Satralizumab	IL-6 receptor antagonist; reduces pro-inflammatory signaling
	Inebilizumab	Anti-CD19 monoclonal antibody; depletes B cells
	Rituximab	Anti-CD20 monoclonal antibody; depletes B cells
	IVIg	Modulates Fc receptor activation; reduces inflammation
	PLEX	Removes pathogenic autoantibodies and immune complexes
	CSF1R inhibitors	CSF1R blockade; reduces microglial/macrophage activation

Abbreviations: BTK, Bruton’s tyrosine kinase; CD, cluster of differentiation; CSF1R, colony stimulating factor 1 receptor; IL, interleukin; IVIg, intravenous immunoglobulin; MS, multiple sclerosis; NMO, neuromyelitis optica; OPC, oligodendroglia precursor protein; PLEX, plasma exchange; S1P, sphingosine-1-phosphate.

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
