# Peer review of "Microglia/Macrophages in Autoimmune Demyelinating Encephalomyelitis (Multiple Sclerosis/Neuromyelitis Optica)"

_ijms, 2025, doi:10.3390/ijms26083585_

Round 1
Reviewer 1 Report
Comments and Suggestions for Authors
In the manuscript entitle "Microglia/Macrophages in Autoimmune Demyelinating Encephalomyelitis (Multiple Sclerosis/Neuromyelitis Optica)",
- it focused on microglia and macrophages in autoimmune demyelinating diseases like multiple sclerosis and neuromyelitis optica. It dicussed on their distinct origins, different roles in these diseases, and emphasizes the latest research progress in this area.
- The overall structure of the article is clear with well, however, to make it more catering to the general reader interest, some revisions are probably needed.
- The readers of the journal International Journal of Molecular Sciences is coming from non-medical background, can the author provide more background knowledge of MS and NMOSD? a figure showing the basic symptoms would benefit the readers.
- While the flow between major sections is generally smooth, there are a few places where transitions between subsections could be enhanced. For example, line 248, the title for section 5 should be a brief description, instead of a short abbreviation.
- Although there is only one figure in the manuscript, the figure should be improved to make it more informative.
- The figure legend should be a brief description of the figure rather than a short paragraph stating the roles of microglia and macrophages in MS and NMO.
- The current version is confusing for a reader which is not an expert in the field, I suggest the author to better to present the statements, for example what does the arrow mean? What does the color of each section stand for?
- Overall, I hope these revisions would help to make the manuscript more accessible for a broader audience while maintaining its scientific rigor.
Please improve the flow of the manuscript, to make it more informative.
Author Response
Reviewer 1
Comments and Suggestions for Authors
In the manuscript entitle "Microglia/Macrophages in Autoimmune Demyelinating Encephalomyelitis (Multiple Sclerosis/Neuromyelitis Optica)", it focused on microglia and macrophages in autoimmune demyelinating diseases like multiple sclerosis and neuromyelitis optica. It dicussed on their distinct origins, different roles in these diseases, and emphasizes the latest research progress in this area.
The overall structure of the article is clear with well, however, to make it more catering to the general reader interest, some revisions are probably needed.
The readers of the journal International Journal of Molecular Sciences is coming from non-medical background, can the author provide more background knowledge of MS and NMOSD? a figure showing the basic symptoms would benefit the readers.
Response:
Thank you for this valuable suggestion. In response, I have added a new figure (Figure 1) that summarizes the typical clinical course and symptoms of MS and NMOSD, in order to improve accessibility for readers without a medical background.
While the flow between major sections is generally smooth, there are a few places where transitions between subsections could be enhanced. For example, line 248, the title for section 5 should be a brief description, instead of a short abbreviation.
Response:
I appreciate your insight. I have revised the titles of Sections 5 and 6 to read “Multiple Sclerosis: Disease Mechanisms and Microglia/Macrophage Involvement” and “Multiple Sclerosis: Disease Mechanisms and Microglia/Macrophage Involvement” for clarity and to better reflect their content.
Although there is only one figure in the manuscript, the figure should be improved to make it more informative.
The figure legend should be a brief description of the figure rather than a short paragraph stating the roles of microglia and macrophages in MS and NMO.
The current version is confusing for a reader which is not an expert in the field, I suggest the author to better to present the statements, for example what does the arrow mean? What does the color of each section stand for?
Response:
I appreciate your comments regarding the original figure. I have replaced it with a redesigned and expanded version (now Figure 2), which provides a more structured and visually intuitive comparison of microglial and macrophage functions in MS and NMOSD. The background color (green for microglia, red for macrophages) and text color (red for pro-inflammatory, blue for reparative functions) are now clearly defined in the revised figure legend. I believe that this new figure significantly improves the clarity and readability for a broader audience.
Comments on the Quality of English Language
Please improve the flow of the manuscript, to make it more informative.
Thank you for the suggestion. I have had the manuscript revised by an English editing service (Edanz).
Reviewer 2 Report
Comments and Suggestions for Authors
Manuscript ijms-3507719-v1, Review "Microglia/Macrophages in Autoimmune Demyelinating Encephalomyelitis (Multiple Sclerosis/Neuromyelitis Optica) by Ryo Yamasaki.
In this review manuscript, the author is interested in the role of microglia and macrophages in autoimmune demyelinating diseases such as multiple sclerosis and neuromyelitis optica spectrum disorder.
This review first describes the origins of microglia and macrophages with their surface markers and their recent classification. Then, the author describes their mechanisms of activation, including transcriptional regulation, the regulation of their phagocytic properties, and their role in disease and repair. After a brief description of their interactions with other neural cells, the authors discuss about the therapeutic potential of targeting microglial and macrophage activity in multiple sclerosis (MS) and neuromyelitis optica spectrum disorder (NMOSD), including insights gained from animal models. Finally, the author discusses about similarities and differences in microglial and macrophage actions in these diseases.
This review manuscript is of interest for the reader. However,it would benefit from a more extensive description of the mechanism recruiting/activating microglia vs. macrophages in these diseases.
In particular, Point 7.1.2. BBB Disruption, and Point 7.2.4. Recruitment of Peripheral Macrophages should be largely extended in order to document in more details their mechanisms of pathogenicity.
Author Response
Reviewer 2
Comments and Suggestions for Authors
Manuscript ijms-3507719-v1, Review "Microglia/Macrophages in Autoimmune Demyelinating Encephalomyelitis (Multiple Sclerosis/Neuromyelitis Optica) by Ryo Yamasaki.
In this review manuscript, the author is interested in the role of microglia and macrophages in autoimmune demyelinating diseases such as multiple sclerosis and neuromyelitis optica spectrum disorder.
This review first describes the origins of microglia and macrophages with their surface markers and their recent classification. Then, the author describes their mechanisms of activation, including transcriptional regulation, the regulation of their phagocytic properties, and their role in disease and repair. After a brief description of their interactions with other neural cells, the authors discuss about the therapeutic potential of targeting microglial and macrophage activity in multiple sclerosis (MS) and neuromyelitis optica spectrum disorder (NMOSD), including insights gained from animal models. Finally, the author discusses about similarities and differences in microglial and macrophage actions in these diseases.
This review manuscript is of interest for the reader. However, it would benefit from a more extensive description of the mechanism recruiting/activating microglia vs. macrophages in these diseases. In particular, Point 7.1.2. BBB Disruption, and Point 7.2.4. Recruitment of Peripheral Macrophages should be largely extended in order to document in more details their mechanisms of pathogenicity.
I appreciate this recommendation. In the revised manuscript, I have expanded Sections 7.1.2 and 7.2.4 to provide detailed mechanisms of blood–brain barrier (BBB) disruption and peripheral macrophage recruitment in MS and NMOSD.
In Section 7.1.2, I now include the roles of MMPs, ROS, VEGF, and pro-inflammatory cytokines in BBB disruption, citing recent literature (e.g., Lopes Pinheiro et al., 2016).
In Section 7.2.4, I have added a comparative explanation of chemokine signaling pathways (e.g., CCL2, CCL5, CXCL10 in MS; CCL2 and C5a in NMOSD), and described how these molecules contribute to the differential recruitment and phenotypic polarization of peripheral macrophages in each disease context.
Round 2
Reviewer 1 Report
Comments and Suggestions for Authors The author has addressed my commenets, and the paper can be accepted.